# Design, Fabrication, and Characterization of Inkjet-Printed Organic Piezoresistive Tactile Sensor on Flexible Substrate

**DOI:** 10.3390/s23198280

**Published:** 2023-10-06

**Authors:** Olalekan O. Olowo, Bryan Harris, Daniel Sills, Ruoshi Zhang, Andriy Sherehiy, Alireza Tofangchi, Danming Wei, Dan O. Popa

**Affiliations:** Louisville Automation & Robotics Research Institute, University of Louisville, Louisville, KY 40208, USA; bryan.harris.1@louisville.edu (B.H.); daniel.sills@louisville.edu (D.S.); ruoshi.zhang@louisville.edu (R.Z.); andriy.sherehiy@louisville.edu (A.S.); alireza.tofangchi@louisville.edu (A.T.); danming.wei@louisville.edu (D.W.); dan.popa@louisville.edu (D.O.P.)

**Keywords:** tactile sensor, strain gauge, aerosol jet printing, additive manufacturing, PEDOT:PSS

## Abstract

In this paper, we propose a novel tactile sensor with a “fingerprint” design, named due to its spiral shape and dimensions of 3.80 mm × 3.80 mm. The sensor is duplicated in a four-by-four array containing 16 tactile sensors to form a “SkinCell” pad of approximately 45 mm by 29 mm. The SkinCell was fabricated using a custom-built microfabrication platform called the NeXus which contains additive deposition tools and several robotic systems. We used the NeXus’ six-degrees-of-freedom robotic platform with two different inkjet printers to deposit a conductive silver ink sensor electrode as well as the organic piezoresistive polymer PEDOT:PSS-Poly (3,4-ethylene dioxythiophene)-poly(styrene sulfonate) of our tactile sensor. Printing deposition profiles of 100-micron- and 250-micron-thick layers were measured using microscopy. The resulting structure was sintered in an oven and laminated. The lamination consisted of two different sensor sheets placed back-to-back to create a half-Wheatstone-bridge configuration, doubling the sensitivity and accomplishing temperature compensation. The resulting sensor array was then sandwiched between two layers of silicone elastomer that had protrusions and inner cavities to concentrate stresses and strains and increase the detection resolution. Furthermore, the tactile sensor was characterized under static and dynamic force loading. Over 180,000 cycles of indentation were conducted to establish its durability and repeatability. The results demonstrate that the SkinCell has an average spatial resolution of 0.827 mm, an average sensitivity of 0.328 mΩ/Ω/N, expressed as the change in resistance per force in Newtons, an average sensitivity of 1.795 µV/N at a loading pressure of 2.365 PSI, and a dynamic response time constant of 63 ms which make it suitable for both large area skins and fingertip human–robot interaction applications.

## 1. Introduction

### 1.1. Tactile Sensing for Robotics

Tactile sensors which measure the force and pressure of contact during a physical interaction with a robot have been studied by many researchers over the last four decades [1,2]. A leading area of application for these sensors comprises robot “skins” that can react to touch during physical interaction in a similar manner to human skin. Human skin is equipped with numerous neuro-sensory pathways that enable us to directly perceive our surroundings through physical contact, including an awareness of touch, heat, and vibration, transmitting the perceived signals directly to the brain. If placed on a robot, such sensors can deliver touch information to the robot not just at the end effector but also across the whole body in order to enhance safety, navigation, and the comprehension of human intent and mobility [3]. In previous research, a variety of such sensors were proposed to monitor pressure, force, and torque [4]. However, due to many remaining challenges in their usability, reliability, and measurement sensitivity, the study of robot skins has continued in both lab and industrial environments. Lumelsky, Shur, and Wagner were among the earliest who proposed the concept of a large-area, flexible array of InfraRed sensors with data-processing capabilities which they call a “sensitive skin”, which covers the entire surface of a robot [1].

Other notable early works which developed whole-body robotic skins used organic thin-film transistors (OTFTs) and a-Si:H TFTs fabricated on flexible plastic substrates at the University of Tokyo [5] and polyvinylidene fluoride (PVDF) tactile sensors at Tohoku University [6]. Later, modular robotic skin designs, such as ROBOSKIN [7] and HEXOSKIN [8], were custom-developed by academic robotics labs in Italy and Germany. These modules contained not just capacitive tactile sensor elements but also electronics for signal acquisition, conditioning, and networking to the robot controller. Concurrently, numerous pressure-sensitive arrays have been commercialized; however, these products are mainly applied in biomechanics rather than robotics. Perhaps the most “integrated” skin–robot effort to date has been directed toward sensitive surfaces for manipulation with robotic hands and fingertips. Examples of commercially available sensors are the Flexiforce [9] and the BioTac, the latter of which provides finger “modules” for commercial robotic hands [10]. The latter can detect heat, vibration (for texture), and pressure, sensing using a combination of thermistors and fluidic pressure sensors with a spatial resolution of 3 mm, a vibration bandwidth of 1 KHz, a thermal/pressure bandwidth of 50 Hz, a dynamical range of 30 mN–30 N, and less than 5% sensitivity/hysteresis. While these sensors have been tuned for installation on fingers, it is difficult to scale them to other parts of the robot and to make them cost-effective for home use or rugged enough to be subjected to daily “wear and tear” in a general human–robot interaction scenario.

### 1.2. Sensor Requirements and Transduction Types

Dahiya et al. [11] discussed the performance required for tactile sensors to achieve a level of sensitivity similar to human skin. The compactness of the tactile sensor’s design resolution and its improved manufacturability and maintainability are some of the desired outcomes of any proposed tactile structure. Their study of tactile “skins” focused on the choice of materials, ideal fabrication methods, sensing mediums, packaging, and associated electronic systems in order to analyze sensory feedback. Compared to other flexible polymers, Kapton, which is also commonly referred to as polyimide for flexible electronics, is a good substrate choice because of its flexibility, chemical inertness, and capacity to endure high temperatures.

A human interaction consisting of a typical light stroke will have a pressure of about 0.3 Pa (30 g/cm^2^), while a push or slap will have a pressure of more than 10 Pa (1000 g/cm^2^) [12]. Furthermore, these sensors must be integrated into the robot and be able to process large amounts of data quickly enough so that the robot can make use of it in a feedback loop. Several modes of operation have been explored via varying modes of transduction for detecting, qualifying, and converting these analog signals into discreet quantities used for actuating mechanisms. Some accessible sensing media include piezoresistive or piezoelectric actuation techniques and resistive, inductive, capacitive, and optical fiber sensing.

References [13,14,15,16,17] present tactile sensors with piezoresistive modes of transduction which experience electrical changes in response to external stimuli with high gauge factors (GF > 100) and high spatial resolutions suitable for fingertip applications within a range of 0.25–1 mm. However, they are non-flexible or cannot be conformed to the 3D surfaces often found on robots. Capacitive sensing mediums detect forces via alterations in the relative static permittivity of dielectric layers between substrates [18,19,20,21]. Although they are very highly sensitive, compatible with static force measurements, power-efficient, and easily configured in sensor arrays of several orders in comparison to other sensing media, they are often susceptible to interference from approaching objects which change the fringe field of the capacitor [22,23,24,25,26]. This phenomenon can lead to ambiguous signals during measurement and significant hysteresis. A piezoelectric sensing medium for tactile sensing, as described in [27,28,29], generates a voltage response from applied mechanical stress. However, when an external force is applied, the structure deforms. In this situation, the charge centers of the anions and cations split and create electric dipoles, causing a piezo-potential to arise. As a result, the external circuit is forced to conduct a flow of free electrons in order to screen the piezo-potential and establish a new balanced state. Qijun et al. presented a medium described as a piezo-potential-powered graphene transistor with gauge factor of 389 and good durability over 3000 bending and releasing cycles. A significant challenge in the use of piezoelectric sensors is their sensitivity to temperature [30,31,32,33,34].

Other types of transducers include optical tactile sensing, described in [35,36,37], which employs the change in light intensity in media with different refractive indices, and triboelectric sensing, which uses the electrostatic charge created when two distinct materials come into contact before separating. This charge results from the transfer of electrons due to the frictional forces that arise during contact and separation [38,39,40].

### 1.3. Contributions

In our previous work [41,42,43], tactile sensors were fabricated using lengthy processes in a cleanroom. The substitution of the cleanroom process with additive manufacturing was proposed in [44,45] by depositing the organic polymer required for its piezoresistive behavior via inkjet printing and substituting the spin-coating technique used with PEDOT:PSS poly (3,4-ethylene dioxythiophene)-poly(styrene sulfonate) with aerosol jet printing and ink-jetting instruments. The results showed a significant reduction in fabrication time and improvements in yield and sensitivity. Other past research reported on the design and fabrication of single tactile strain gauges, using the principles of aerodynamics to precisely align atomized ink droplets on flexible substrates in the NeXus for sensor electrodes [46,47,48,49]. The tactile structures designed and fabricated are ideal for eliminating directional effect due to their circular topology, as characterized in [41,46], but become significantly less effective in a smaller, compact strain gauge size below 6 mm in diameter.

In this work, a new sensor design is proposed which imitates a human fingerprint to fulfill the design criteria in [41] of an increased spatial resolution achieved via the use of the inkjet printers in the NeXus microfabrication platform. As a result, our new sensors are half the size of what was previously possible. This completely substitutes the entire photolithographic fabrication process in the cleanroom with aerosol jet printing that can deposit water-based or solvent-based ink droplets with a viscosity of up to 30 cp for extruding silver printed lines as a sensor electrode and an ink-jetting instrument for depositing PEDOT:PSS, which is the base sensing organic polymer responsible for the piezoresistive nature of the tactile sensor. The conductive polymer composite called PEDOT:PSS has a property called piezoresistivity which is a complex phenomenon in which electrical resistance varies in response to mechanical deformation. This effect happens as a result of changes in the spacing and alignment of the conducting polymer chains (PEDOTs) when the material is stressed or deformed, such as when it is bent or compressed. Consequently, the material’s electrical conductivity is impacted. The electrical resistance increases when chains are pulled apart and lowers when they are forced closer together. The use of PEDOT:PSS in piezoresistive sensors is based on this shift in resistance. A second effect is related to the changes in the electronic structures of PEDOT and PSS due to the mechanical deformation, affecting the polymer chains, the bonding distances between atoms, and subsequently the charge distribution. It can be speculated that this causes changes in the energy difference between occupied and unoccupied electron states in PEDOT and PSS. Analogous to the piezoresistive effect, in bulk solid semiconducting materials, in addition to the geometrical (volume) part affecting resistivity there is also a component related to changes in the electron structure of the material, inducing changes in the band gap value (for example in the case of SI). According to research, PEDOT:PSS has beneficial qualities such a high gauge factor, flexibility, and good process compatibility. According to reports, as a sensing material, PEDOT:PSS has potential gauge factors ranging from 6.9 to 17.8 [50,51]. In this paper, we describe fabrication procedures, including curing samples in an oven and the use of the four-point probe method to assess the performance of the printed sensors. As discussed in previous publications [41,44], the organic polymer PEDOT:PSS undergoes a preparation process involving the mixture of solvents and compounds such as DMSO (dimethyl sulfoxide) and PVP (polyvinylpyrrolidone) to improve its wettability and reduce its viscosity, making it applicable for spin coating which is not necessary with the use of the inkjet printers within the NeXus. In view of this, a suitable PEDOT:PSS stock (Clevios PH 1000) which is adaptable to the direct-writing jetting technique was obtained from Heraeus, thus eliminating the need for the preparation of a PEDOT:PSS recipe and further reducing the processing time.

The tactile sensor described in this paper increases the possibility of obtaining a higher spatial resolution due to its small and compact structure, which increases its ability to be condensed in given area when compared to [47]. Finally, to establish a dynamic relationship between the tactile sensory feedback and the indented force measured, we modeled the tactile feedback response data using the system identification framework. We also indented the SkinCell array in 29 points by scanning the sample under a force load applicator tip which was custom-developed in our lab. Using the collected spatial data, we interpolated a Gaussian parametric response model for the array and proposed two novel methods for evaluating the spatial resolution of a SkinCell. The results demonstrate that the SkinCell has an average spatial resolution of 0.827 mm, an average sensitivity of 0.328 mΩ/Ω/N, expressed as the change in resistance per force in Newtons, an average sensitivity of 1.795 µV/N at 2.365 PSI, and a dynamic response time constant of 63 ms.

The article is organized as follows: Section 2 discusses the design, fabrication, and sintering process of the fingerprint tactile structure, and Section 3 describes the experimental setup and electronics used to test the tactile sensor array and its durability. Section 4 presents the results and the system identification of the tactile sensor, and Section 5 concludes the work and discusses future work.

## 2. Design of Fingerprint Tactile Sensors

The proposed novel tactile sensor design, inspired by the shape of a human fingerprint (Figure 1), was based on a performance simulation study of a miniaturized and compact spatial structure. The design of our sensor has two main elements: (1) planar metallic, miniature electrodes and (2) a piezoresistive PEDOT:PSS thin film layer. The metallic electrode structure with a nominal trace width of 60 microns was manufactured using an aerosol jet printing (AJP) technique and silver nanoparticle ink, using the critical part of the sensor’s AJP electrodes to estimate the conductivity of the design with respect to realistic conditions during printing. For that purpose, Equation (1), shown below, can be used to derive the conductance of the Ag ink lines of our sensors printed via AJP:(1)$$G=\sigma \frac{TW}{L},$$
where the conductivity *σ* of the NovaCentrix^®^ silver ink is 7.05 × 10^6^ S/m (a manufacturer-reported value), which is comparable to the conductivity of bulk silver–6.3 × 10^6^ S/m; the length of the fingerprint tactile structure printed on the flexible substrate is *L* = 55.26 mm [52]; T is the thickness of the deposited silver ink, measured at 2 microns; and W is the width of the conductive silver line, measured to be 60 µm. The tactile sensor’s design is depicted in Figure 1A with a diameter of 3.80 mm. The resulting conductance of the ink lines of the tactile sensor was approximately 1.53 × 10^−2^ S. Figure 1B, on the other hand, shows the computer-aided design of a four-by-four fingerprint structure design which replicates a single tactile structure. The structure, which has an overall length of 45.16 mm and a width of 28.98 mm, is referred to as a SkinCell and consists of 16 identical sensors, all connected with a common ground and individually patterned electrodes to corresponding connectors. These parameters were chosen in accordance with the COMSOL simulation in [41], which highlighted the significance of designs with improved spatial densities and features that allow for flexibility in defining various sensor structures and resolutions.

The sensor patches’ electrical connectors (pins) are located on opposite sides of the sensor patch (Figure 1C, top/bottom), accommodating eight pins each for signal and one pin for the common ground. The empty box required on each side is situated somewhat as alignment mark to not only properly align and facilitate the back-to-back assembly of two closely matched sensor patches but also to complete the half-Wheatstone-bridge configuration of a functional pair. These paired individual sensors, each referred to as a “tactel”, send signal responses to the connectors, transferring external stimuli through the connectors to the electronics testing setup for data acquisition. The fingerprint sensor array design uses inkjet printing technology and can be printed at a constant substrate velocity, resulting in uniform line widths and a fabrication yield of nearly 100% when compared to other models fabricated in the cleanroom [49]. In comparison to previous models fabricated using cleanroom methods, our design is more durable to wear and tear and resists peeling off in when contact with electronic ZIF connectors. Furthermore, a spiral geometry utilizes the space on the substrate in a very compact manner, basically making very compact interdigitated electrodes (IDEs) with a circular symmetry. In comparison with other IDE electrode designs, such as the one in [53], the sensitivity is improved by a factor of 3.

## 3. Fabrication of Fingerprint Tactile Sensors

### 3.1. Aerosol Jet Printing System

To carry out the fabrication process presented in the flow chart below (Figure 2), a unique robotic system, the NeXus, was used for the realization of multiscale manufacturing and rapid prototyping. The Nexus is a custom robotic platform designed and developed at LARRI, which combines automated assembly and additive manufacturing. It integrates a number of different subsystems, such as a custom 6-DOF positioner, an Optomec Decathlon Aerosol Inkjet printer, a 3D FDM printer, a Nordson EFD PicoPulse^®^ inkjet station, and an intense pulse light (IPL) sintering station from Xenon Corp. [54]. Along with an e-textile loom/weaving device, a micro-assembly/inspection station and an industrial robot tool changer for various automated operations are additional features. The SkinCell tactile sensor structure was printed by the NeXus system using an OPTOMEC^®^ aerosol inkjet printer on a flexible Kapton^®^ substrate. PEDOT:PSS, an organic polymer responsible for the piezoresistive phenomenon of the tactile sensor, was then deposited using a PicoPulse^®^ device. A custom 6-DOF positioner in the NeXus was created and incorporates a long linear stage and five high-precision motorized stages to transport the substrate to the appropriate stations for printing the sensor electrode and the deposition of PEDOT:PSS. The precision motorized stages ensure XYZ displacement with 1 micron resolution, as well as sample rotation and tilt, while the long linear stage transports the sample between various printing, curing, and metrology instruments. In the Nexus, the substrate is calibrated and aligned prior to the aerosol and inkjet printing processes. This allows for accurate motion control and the deposition of ink in the appropriate place. The printing process begins after the alignment is finished, and the structure is plasma-treated after the sensor electrode is printed and cured. The plasma treated sample is transferred back to the Pico Pulse station at Nexus for deposition of PEDOT:PSS. The substrate (which is either an FPC Kapton for a single-sensor fabrication or a blank Kapton sheet for the fabrication of a four-by-four sensor patch, as shown in Figure 3A) is secured on the 6-DOF positioner’s sample stage, as shown in Figure 3B, and lined up with the print head for printing. According to the process parameters described in Table 1, the Nordson EFD Pico Pulse^®^ is used for depositing the organic polymer responsible for the piezoresistive behavior of the tactile sensor, as shown in Figure 3F. This ink-jetting instrument has a controller that actuates the piezoelectric actuator print head to which a 50-micron nozzle and fluid syringe are attached. In this study, the controller was tuned to the parameters shown in Table 2 to produce an evenly overlapping overlay of PEDOT:PSS [54]. The following dispensing parameters were adjusted during the print process:
-The waveform times (open/close/pulse);-The cycle of the ink-drop dispensing period corresponding to the dispensing frequency, $f_{d}$ (1–250 Hz);-The stroke force of ink droplet ejection, expressed in % for which the maximum force equals 100%;-The deposit height, h (3 mm), which is the distance from the print head nozzle to the substrate;-The temperature of the ink, $T_{f}$ (40 °C);-The air pressure, *P_a_* (20 psi), which is the fluid pressure in the print head valve assembly.

### 3.2. Tactile Sensor’s Aersol Jet Inkjet Printing Trajectory

For the inkjet systems, motion control and ink deposition control were realized through the LabView control interface of the NeXus system. The conductive fingerprint design, printed via the Optomec using NovaCentrix^®^ JS-A426 silver ink, was realized through Line-Arc Trajectories, a machine control language unique to the Newport controllers built into the NeXus system that control the 6-DOF positioner. These trajectory files enable the printing of a continuous line at a constant velocity, producing smooth, curved features and a constant line width throughout the structure. The PEDOT:PSS thin film, realized via the Pico Pulse^®^, was controlled through an integrated G-code parser built into the LabView control interface. Through this parser, both deposition control and motion control were dictated. For the deposition of the thin film, an overlapping serpentine structure was used to efficiently and reliably deposit the PEDOT:PSS ink into a film on top of the conductive structure.

### 3.3. Tactile Sensor Sintering and Plasma Treatment

To remove the solvent and increase the compactness of the silver nanoparticles, the Ag ink must be sintered after printing. The particles combine and adhere during the curing process, increasing the conductivity of the printed electrodes. The curing procedure was carried out using a thermal scientific Lindberg vacuum oven, as shown in Figure 3C. The sensor’s substrate with the printed lines was cured in the oven for 20 h at a temperature of 200 °C [45]. An oven-cured sample is depicted in Figure 3D.

### 3.4. PEDOT:PSS Deposition

At this stage, before the deposition of the PEDOT:PSS, the cured sample should be tested as an open circuit before it is placed in a vacuum plasma chamber, as shown in Figure 3E, in preparation for the deposition of the PEDOT:PSS. The reason for the plasma treatment is to improve the wettability of the Kapton substrate with respect to the PEDOT:PSS ink and consequently improve the quality and uniformity of the printed film. This preprocessing step also provides a means of reducing the thickness of the film [55]. A Harrick^®^ Plasma Cleaner device was used to administer the plasma treatment. The substrate was placed inside the chamber, evacuated, and subjected to air RF plasma at 30 W for two minutes after reaching low vacuum. The tactile sensor patterned substrate was prepared for Pico Pulse^®^ inkjet printing with PEDOT:PSS for an effective time of 30 min [54]; this setup is displayed in Figure 3F. Three layers of PEDOT:PSS thin films were printed on top of the electrode structures with a thickness measured via a Dektak^®^ Profilometer that amounted to approximately 1 micron [55]. Immediately after the deposition step was completed, the sample was placed within the oven, as seen in Figure 3G, for additional curing for a duration of 30–45 min, leaving the PEDOT:PSS effectively attached to the sensor electrode, as seen in Figure 3I. As mentioned earlier, in this work, we studied the behavior of PEDOT:PSS thin films with thicknesses not exceeding 1 micron which corresponded to one, two, and three printed layers. It was observed that the sensor with three layers produced the most reliable results, whereas the single-layer sensor’s behavior was inconsistent. We did not study the response of a sensor with a greater number of layers and a thickness above 1 micron for a given type of PEDOT:PSS ink (from Heraeus) as our goal was to demonstrate our capability to fabricate PEDOT:PSS thin films that were as thin a possible using the inkjet printing method, producing structures with acceptable behavior for the given application and reasonable manufacturing times compared to cleanroom fabrication.

### 3.5. Sensor Patch Lamination

A pair of double-sided pieces with two entirely manufactured and bonded skin sensor patches comprised a functionally constructed robot skin sensor module. Each sensor patch was laminated with Kapton tape to increase the ease of handling, prevent the silver electrode connections from breaking and the PEDOT:PSS from wearing off, and protect the sensor area from ambient moisture. The two fully constructed sensor patches were paired and positioned back-to-back, with the sensor region facing outward during this operation. To account for the temperature drift of the sensors, the sensor patches were attached on both sides. Importantly, the lamination kept the sensor patches fastened well to the connectors on the circuit board, enabling a constant and consistent feedback response. The process of lamination depicted in Figure 4 is described in detail below.

1.Placing the completely fabricated sensor patches:

A square-shaped Kapton sheet with four completely fabricated sensor patches was placed on a flat surface, ready to be processed.

2.Detaching the sensor patches:

The square Kapton sheet was cut to detach the four sensor patches. Two closely matched pairs were chosen for processing based on the measured resistance of the tactile sensors.

3.Protecting the sensor electrode connectors:

To laminate the closely matched pairs, the sensor electrodes were protected by covering them with sticky notes on either side of the sensor patch.

4.Protecting the surface with Kapton tape:

The closely matched sample pair requires careful handling to avoid touching the sensor surface. The surface was covered with Kapton adhesive tape. The Kapton tape covered the sensor regions, leaving the electrode open to make a connection with the circuit for testing.

5.Using double-sided tape for alignment:

One of the matching pairs was turned upside down and taped to the flat surface with double-sided tape in preparation for its alignment with the other patch.

6.Aligning the matching pairs:

With the first patch held down firmly with double-sided tape, the other sensor patch was aligned properly, utilizing the rectangular alignment box at the end of the connectors.

7.Applying adhesive spray to the aligned pair:

The two layers of the double-layer sensor array were separated by wiping paper on the center of the flat substrate on which they were mounted. Next, we evenly sprayed 3M^®^ contact glue between each pair. For back-to-back Kapton alignment and adhesion, a 3M 13.8 oz. Super 77 Multipurpose Spray Adhesive was sprayed evenly in between the pairs. It is designed to withstand a wide temperature range. It typically has good resistance to heat and cold, making it suitable for many applications and providing secure bond.

8.Smoothing the adhered pair:

A double-sided construction was made using the brayer and another wiping paper placed on top of the laminated sensor arrays. The wiping paper was removed after the pair had been closed together and the clip was removed.

9.Curing and trimming the outline:

To cure the adhesive, a laminated double-layer sensor array was placed between two flat substrates, a hefty metal block was placed on top, and the assembly was baked in a standard oven at 75 °C under vacuum for 10 min. The laminated sensor array was then transferred outside, and its contour was cut to make the bottom edge flush with the bottom so that two ZIF connectors could be inserted and connected to our conditioning electrical circuit.

10.Cleaning the laminated paired sensor patch:

The resultant laminated pair was cleaned of any adhesive residue, using acetone and IPA in preparation for the test connection.

### 3.6. Fabrication Duration Analysis

The layered design of a tactile sensor patch, or SkinCell, is depicted in Figure 5. The fabrication is predicated on two main processes: (1) the sensor electrode, which involves the inkjet printing of a conductive metal in a specific design, and (2) the deposition of the organic material responsible for the piezoresistive behavior of the tactile sensors, PEDOT:PSS. In past publications [41,42], we carried out the entire fabrication process in a cleanroom, using sputtering and photolithographic techniques to pattern the sensor electrodes and ensure the deposition of PEDOT:PSS on the sensor region. The drawbacks experienced with the cleanroom process were the high cost of processing, the low level of design adaptability, and the longer time taken when compared to the inkjet printing processes using the NeXus. These factors made it cumbersome, and a high level of precision was required to maintain high yields of working sensors. The lengths of time taken to complete the fabrication process using cleanroom techniques and direct-writing inkjet processes within the NeXus are analyzed below.

1.Sensor Electrodes

For the patterning of sensor electrodes with conductive metals such as gold in the cleanroom, a photolithographic technique for creating the window for metal deposition coupled with metal sputtering and a wet-etching process which lasted approximately “2 h 53 min” were required to pattern the electrodes, while for the inkjet printing process, the metal ink was loaded into the OPTOMEC^®^ Aerosol Jet printer and optimized using the printing parameters specified in Table 1. The approximated print time was “1 h 5 min”.

2.PEDOT:PSS Deposition

For the deposition of PEDOT:PSS, the layer must be formed only on the sensing area to avoid shorting the circuit. To execute this in the cleanroom as described in [41], spin-coating with the PEDOT:PSS was carried out; however, since the PEDOT:PSS covered the entire surface, a series of steps were required to preserve the PEDOT:PSS on only the sensor surface, removing the rest from the unwanted regions. The average time taken to perform the entire process was approximated to be “7 h 24 min”. For the Inkjet printing process within the Nexus, the PEDOT:PSS was loaded in a direct-writing inkjet printing tool called the Pico Pulse^®^, described earlier, depositing the organic polymer precisely on the sensor surfaces in an approximated time of “45 min”.

In summary, the cleanroom process for the fabrication of skin sensor patches requires a total average of “10 h 17 min”, while the use of direct-writing inkjet printing techniques averages a total of “1 h 50 min”, excluding the curing time in the oven, which could vary from 30 min to 15–20 h or few seconds using the IPL (intense pulse light) present within the Nexus fabrication platform. This is indicated below in the pie chart in Figure 6.

## 4. Experimental Test Setup and Results

The experimental test setup was described in our previous studies [41,46,56] and made use of a plunger and a load cell to evaluate how well the sensors of each patch worked when subjected to varied forces. The testbench was controlled via a cRIO-9074 Real-Time controller from National Instruments^®^, and NI9205 and NI9516 devices were installed in the general-purpose I/O ports. For one-way linear motion, the NI9516 units were coupled to Newport^®^ travel stages. The first stage was positioned vertically, and the plunger and load cell were fastened. Testing with force was achieved by modifying the motor’s position in response to feedback from the load cell. The system was connected to an electronic circuit board and a LabVIEW^®^ interface, which automated the testing and evaluation of the tactile sensor. The load cell regulated the force applied to the tactile sensor as the LabVIEW^®^ front panel generated a real-time force load profile.

### 4.1. Spatial Response for a Single Tactel

The characterization of the tactile sensor based on the load applied to varying locations on the geometry of the manufactured tactile fingerprint structure on the Kapton substrate was carried out. As shown in Figure 7, we tested 16 sub-indenter positions at each of the following angles: 0°, 45°, 90°, 135°, 180°, 225°, 270°, and 315°, including the center. Tactile responses to indentations at distances from 5 mm and 10 mm to the center were recorded. Three seconds were spent keeping the indenter in place at every location, with 1 N applied at each sub-indenter location separately measured. Using an Agilent^®^ 34970A and an Agilent^®^ 34901A 20-channel multiplexer, a four-wire resistance measuring approach was used to gather resistances. Figure 8 shows the response of the tactile sensor. Figure 8A presents the variation in the magnitude of the response depending on the location of the point at which the load was applied. As we observed, the response of the sensor to the indentation was the strongest at the center of the tactile sensor, whereas the results from the other locations appear to follow the same trendline, with a reduction in sensitivity when moving away from the center with the exception of the response from an orientation angle of 270°. We assume that this is due to the presence of the printed connecting silver lines beneath the sub-indenter which are impacted by the applied force (Figure 7). The linear and quadratic fits of the sensitivity with respect to the applied force were determined using 30 samples. The mean baseline resistance was recorded as 493 Ω, with a standard deviation of 0.06 Ω. The resulting sensor demonstrated a linear relation between the relative resistance change (|ΔR|/R_0_) and the force F, with a sensitivity of approximately 0.328 mΩ/Ω/N, as shown in Figure 8B. Figure 8C shows the elastic hysteresis of the tactile sensor based on increasing and decreasing loads which was determined within the quadratic fit.

### 4.2. Repeatability of Sensor Measurements

A custom 3D printed reciprocating mechanism was designed, as shown in the schematic in Figure 9a, and manufactured to perform dynamic testing on the soft sensors at a desired frequency and amplitude, as shown in Figure 9b. At its core, the unit utilizes the Scotch yoke mechanism and consists of a sliding track connected to a rod (yoke) that moves back and forth in a straight line, along with a rotating disk (crank) that provided the rotary motion. The yoke is connected to the crank through a pin and bearing which freely moves in the slot. As the crank rotates, the pin pushes the yoke back and forth along the slot, translating the rotary motion into linear motion. Here, the distance of the pin from the center determines the range of motion of the rod(stroke), and the spin speed of the DC motor defines the cycle frequency. One novel feature of this design is that the distance of the pin from the center can be varied from 0.1 mm to 10 mm using a built-in set screw and washer, allowing the unit to operate at a desired amplitude. By obtaining a calibration curve for the speed of the motor vs. the voltage, we were able to operate the machine and test the sensors which were subjected to the desired frequency and amplitude. The unit also has other adjustable screws so the sample may be approached from various heights and to secure the device on the bench for long periods of operation. The sensor pad was placed on a digital scale to monitor the pressing force delivered by the end point of the reciprocating rod. The maximum force applied to the sensor was set at around 100 g at a frequency of 1 Hz.

Figure 9b presents the tactile sensor’s durability profile for a load of about 1 N across a significant number (180,000) of force indentation cycles, for which the change in resistance, ΔR, corresponds to the difference between the baseline and sensor responses. Furthermore, measuring the minimum rolling window, which functions as a high-pass filter for the sample size, eliminates temperature fluctuation effects. As can be seen, there were no significant changes in the ΔR value across 180,000 cycles except for minor fluctuations, clearly displaying the stability and endurance of our sensor. This is especially relevant when comparing our sensor to the brittleness of silicon micromachined sensors [57,58]. Additionally, when compared to the sensor developed by Huang C.Y. et al. [59], it is less stiff and more flexible, allowing it to conform to non-planar surfaces. Finally, it is less expensive.

### 4.3. Resistance and Sensitivity Measurement of Sensor Patch

The resulting sample was sintered in the oven separately for 20 h after the patterning of the silver electrodes and thermally annealed for 30–45 min after the deposition of the PEDOT:PSS. Each sensor’s resistance on the skin sensor patch was measured and recorded after the fabrication process was complete but before the laminating procedure, resulting in a mean resistance of 143 ohms and a standard deviation of 17 ohms. A customized electronics circuit was used in the experiment which is explained in detail in [56]. A fully assembled or laminated skin sensor patch allowed each sensor cell to be placed into a voltage divider circuit (V_sen_), which represented a half-Wheatstone-bridge circuit. The other half of the Wheatstone bridge circuit was completed using a fixed voltage divider circuit and buffered using an operational amplifier (Texas Instruments^®^ OPA1612) voltage follower (V_ref_). V_sen_ and V_ref_ were then amplified using an instrumentation amplifier (INA, Texas Instruments^®^, INA333) to reduce the DC component of the V_sen_ and amplify the force-induced voltage change with a gain of 33. The INA’s output was then digitized via a 24-bit analog-to-digital converter (ADC, Texas Instruments^®^, ADS1258) and collected on a personal computer (PC) via serial. The raw V_sen_ can be calculated via the ADC’s conversion equation and the INA gain. This process is crucial because it selects the sensor patches with very similar readings for pairing. After the selection procedure and resistance measurements were completed, the lamination process was carried out. The resistance values of the two laminated pairs are shown in Table 3. The tops of the sensors experienced an inward compression when strain was immediately applied to the laminated sensor pair using the indenter, whilst the opposite pair of sensors experienced an outward tension. A soft rubber silicone was positioned beneath the sensor patch on the testing station platform, allowing for its deformation as pressure was applied to the sensors [43].

The silicone elastomer under each tactel increased the amount of strain experienced by each SkinCell tactel. A silicone elastomer indenter layer was also placed on top of each tactel to concentrate stress on individual tactels. The bedding design and analysis were presented in detail in another article [43]. Figure 10 shows the sensitivity of the 16 sensors on the patch, indicating the response of the sensors from 0.5 to 2 N. The mean sensitivity of the tactile sensor patch was 1.795 µV/N, with a standard deviation of 0.45 µV/N. It is important to note that the procedure of laminating two tactile sensor patches formed a half-Wheatstone-bridge configuration, cancelling out temperature-induced changes.

### 4.4. Sensor Array Spatial Indentation Results

Figure 11a shows the distributed point of force indentation for the study of the spatial resolution of the sensor patch, using the responses of four tactile sensors located at the center of the sensor patch and based on a force distribution of 2 Newtons applied to the sensors. The force was evenly applied to 29 points distributed along the *x* and *y* axes of the four centered tactile sensors, including the centers of each sensor. In Figure 11b, the circles represent the locations of the 16 tactile sensors on the sensor patch. The Gaussian shape across the four center sensors shows the behavior of the tactile sensors under the influence of the force distribution at the center of the sensors and pressure points away from the center of the sensors. We fit an elliptical Gaussian model to the measured sensor values using a nonlinear least squares fit, specifically using Python’s SciPy implementation of the Levenberg–Marquardt algorithm (“curve_fit”). The Gaussian prediction model for a sensor *i*, *S_i_*(*x*, *y*), depends on the location of the indentation at the *x*, *y* coordinates and can be expressed as follows:(2)$$S_{i}\left(x,y\right)=h_{i}\exp\left(-\frac{{\left(x_{i}-\mu_{x_{i}}\right)}^{2}}{2{\sigma_{x_{i}}}^{2}}-\frac{{\left(y_{i}-\mu_{y_{i}}\right)}^{2}}{2{\sigma_{y_{i}}}^{2}}\right),$$
where $h_{i}$ is the height of the Gaussian peak for a sensor *i* centered at ($\mu_{x}$, $\mu_{y}$), with a standard deviation width of $\sigma_{x}$ (in the *x*-direction) and $\sigma_{y}$ (in the *y*-direction).
(3)$$R^{2}=1-\frac{RSS}{TSS}$$

The quality of fit was determined via the coefficient of determination, denoted by $R^{2}$, ranging from 0 to 1, according to Equation (3), where the *RSS* is the sum of the squares of the residuals and *TSS* is the total sum of the squares of the individual measurements. $R^{2}$ shows how well the measured data fit the Gaussian model and how to predict a measurement given this model [60]. The closer the value of $R^{2}$ is to 1, the greater the ability of the model to explain all the variability in the response data around its mean.

For the four tactile sensors in the center of the SkinCell in this study, we were able to estimate a very good coefficient of determination, as shown in Table 4, along with the Gaussian model parameters of the four central sensors in the array and the offset of the peak of each sensor’s Gaussian from the peak of the measured response.

### 4.5. Spatial Resolution Estimation for Sensor Array

Using the interpolated Gaussian model, we estimated the spatial resolution of the sensor array. Specifically, given an indentation with a known force at a particular target location, the *x* and *y* coordinates of the indentation point can be estimated from the sensor responses measured at the four adjacent corner tactels. The difference between the actual load application point and the estimated point is defined as the spatial resolution of our sensor array. We proposed and evaluated two estimation methods for determining the load application point, namely the weighted averaging method and the elliptical intercept predicted by the model, as described in this Section.

#### 4.5.1. Weighted Averaging Method

This method implements a “weighted averaging” technique similar to the ones proposed in [61,62]. The fundamental concept of this approach is to combine the measured response at the centers of the Gaussian distributions at each of the four corner sensors adjacent to the (*x*, *y*) coordinate. The weights of each contribution are the measured responses $S_{i}$; we can then compute the estimated indented location as the weighted average of the corner locations ($\mu_{xi}$, $\mu_{yi}$) of each sensor’s Gaussian distribution, shown in Figure 10, according to the following:(4)$$x=\frac{\sum \left(S_{i}\mu_{xi}\right)}{\sum S_{i}}$$
(5)$$y=\frac{\sum \left(S_{i}\mu_{yi}\right)}{\sum S_{i}}$$

After total of 20,000 iterations of randomly picking load application points taken at two separate times of 10,000 iterations each, the *x* and *y* coordinates of the estimated weighted averaged method for determining these unknown indentations had some points which were precisely accurate to the target locations, while some were off at certain distances from the target locations. The mean distance of the estimated coordinates for the two separate iterations appears to be the same at 1.1 mm off the target location. Using 13 of the already known indented locations from the measured data, the estimated location was calculated using the weighted average technique, as shown in Figure 12A. It can be seen that 6 out of 13 of the estimated coordinates specified by the smaller circles, “°”, align with the indentation points in the crosses “+”, while the rest were of varying distances away from the mark.

#### 4.5.2. Elliptical Intercept Method

This method uses the Gaussian model equations to estimate the *x* and *y* coordinates of an unknown force indentation. As a result, it is more precise but also more complicated. To solve for the precise coordinate of a force indentation, the intersections of the ellipses representing the responses of the four sensors are used. The Gaussian model is inversed to mirror the formular of a shifted ellipse with different widths and depths, representing the response to each sensor. This is expressed below, derived for each sensor from Equation (2) for some given modeled response *Si*:(6)$$\frac{{\sigma_{yi}}^{2}{\left(x-\mu_{xi}\right)}^{2}+{\sigma_{xi}}^{2}{\left(y-\mu_{yi}\right)}^{2}}{2{\sigma_{xi}}^{2}{\sigma_{yi}}^{2}}=\ln\frac{h_{i}}{S_{i}}$$
(7)$${\sigma_{yi}}^{2}{\left(x-\mu_{xi}\right)}^{2}+{\sigma_{xi}}^{2}{\left(y-\mu_{yi}\right)}^{2}=2{\sigma_{xi}}^{2}{\sigma_{yi}}^{2}\ln\frac{h_{i}}{S_{i}}$$

Let $C_{i}$ = $2{\sigma_{xi}}^{2}{\sigma_{yi}}^{2}\ln\frac{h_{i}}{S_{i}}$, then
(8)$$\frac{{\left(x-\mu_{xi}\right)}^{2}}{\frac{c}{{\sigma_{yi}}^{2}}}+\frac{{\left(y-\mu_{yi}\right)}^{2}}{\frac{c}{{\sigma_{xi}}^{2}}}=1.$$

Since the width of an ellipse centered at ($\mu_{xi},\;\mu_{yi})$ is defined by “2a” and the depth is defined as “2b”
(9)$$\mathrm{Width}\;(w_{i})=\frac{2\sqrt{c}}{{\sigma_{yi}}^{2}}$$
(10)$$\mathrm{Depth}\;(d_{i})=\frac{2\sqrt{c}}{{\sigma_{yi}}^{2}}$$

To determine indented location at *x* and *y*, the equation is expressed and simplified as the formula of a shifted ellipse for the four sensors *i* = 1, 2, 3, and 4.
(11)x−μxi2(wi2)2+y−μyi2(di2)2=1

This gives rise to four elliptical equations with two unknowns (*x*, *y*).
(12)$${d_{i}}^{2}{\left(x-\mu_{xi}\right)}^{2}+{w_{i}}^{2}{\left(y-\mu_{yi}\right)}^{2}=\frac{1}{4}{w_{i}}^{2}{d_{i}}^{2}$$

We found the intersection of these four ellipses (*x*, *y*), using their equations and known values (Table 4) via Python’s SymPy symbolic solver. In Figure 12B, we present an example of one iteration. The predicted estimated location is denoted by the symbol “×”, which is perfectly aligned with the “+” symbol representing the unknown indented location. Meanwhile, the “°” shape indicates the weighted average, estimated to be at a certain distance from the indented location. To verify the accuracy of the elliptical intersect method, we used the location of the known and measured intended force location in Figure 12A and estimated it using the elliptical intersect of the nearby four sensors. The results show an exact match for all locations compared to the weighted average method. Considering that the force indenter used in this study had a 3.86 mm diameter, the weighted average technique falls within the indenter’s sphere of influence. The mean distance from the precise target location was 1.1 mm, which was less than the indentation tip’s radius of 1.93 mm. Consequently, the weighted average method proved to be faster at estimating force-indented locations. The spatial resolution was defined by the difference between the peak of the Gaussian fit and the peak of the measured response, and it equaled 827 microns.

### 4.6. Dynamic Response Characterization of Tactile Sensors

To understand the dynamical responses of our sensors, we employed system identification techniques to define the mathematical relationships between the sensors’ responses and the changing input force signals. As a finite element analysis (FEA) takes a long time to complete and the outputs of the models occasionally do not converge, forcing the simplification of the initial design, system identification modeling of the tactile sensor can be employed as an alternative. This might provide us with an observable trend on its own, but in contrast to a model for system identification, it cannot provide precise insights into the description of the model.

To identify the dynamical responses of our sensors, we used the System Identification Toolbox in MATLAB^®^, using a sampling procedure to obtain synchronized input and output data. The response of the system, assumed to be approximately linear, can be represented using a variety of model structures, including input–output polynomial models, transfer functions, and autoregressive models. In order to evaluate the observed output response in resistance with respect to the force input signal for tactile sensors, a linear time-invariant system, an auto-regressor with an eXtra input model, was used [46]. As seen in Figure 13, a discrete step ladder profile represents the applied force input and the sensor response of a single tactile sensor on a flexible printed substrate. The performance of the sensor was evaluated using a force step ladder profile ranging from 0.5 to 2 Newtons, and the results shown in Figure 13A,B display a negative inversion in the sensor’s response. It can be seen that the sensor’s response was inverted negatively, clearly depicting that the resistance of the sensor and specifically of the PEDOT:PSS layer was reduced when the applied force increased. The discovered model is represented by a state-space model with a predetermined number of states, starting at one and progressively increasing to three. The continuous-time state-space model identified can be represented by the mathematical expression below:(13)$$dx/dt=Ax\left(t\right)+Bu\left(t\right)+Ke\left(t\right)$$
(14)$$y\left(t\right)=Cx\left(t\right)+Du\left(t\right)+e\left(t\right)$$

The process of system identification heavily depends on measurement data, and the identified model was subsequently evaluated for correctness and validity using model validation procedures. This means that a portion of the data were used for the identification of the unknown parameters, and the rest of the data were used for validation.

Three states were added in the third-order model, resulting in a 3 × 3 matrix (A) that characterized the system dynamics, a 3 × 1 matrix (B) that mapped input to state, a 1 × 3 matrix (C) that mapped state to output, and a scalar (D) that mapped input to output. Bias terms or system disturbances were represented by an additional 3 × 1 vector (K). The need to take into account any intermediate states in the system, such as mechanical deformation or temperature changes brought on due to the force applied to the sensor, led to the use of a second-order model. A more thorough knowledge of the system’s dynamics and the tactile sensor’s reactions to diverse force inputs was made possible via the second-order model.

We also formulated a first-order state-space model that only took into account one state variable as a means of comparison. A condensed view of the system’s dynamics was provided by the first-order model. The state could indicate an instantaneous deformation or a change in the sensor as a result of the applied force, suggesting a direct relationship between the applied force and the resulting resistance.

Using the state-space model in continuous time, we were able to produce the model with the best estimated data fit, implementing third-order and first-order state models. This produced an estimated data fit values of 94.9%, 86.75%, and 95.58%, respectively. The models were validated using a validation data set from the system, producing simulated validation data fit values of 65.1%, 63.02%, and 63.92% for both the third-order, second-order, and first-order models, respectively, as shown in Figure 14. The continuous-time state-space models identified for both the second- and first-order models are shown below. The results indicate that the first-order model has a good validation data fit compared to the others, and it is also the simplest; therefore, we selected it. This model reveals that the tactel response had a time constant of approximately 63 ms, indicating great performance for human–robot interaction.

Fitted third-order state-space model:
AB**K****C****D**−4.116 × $10^{-7}$
8.157 × $10^{-5}$
−1.141 × $10^{-6}$
2.05 × $10^{-5}$
−2.261 × $10^{-4}$
−24940.1928−0.01798−0.09251−0.133−14.91.485−2.828−7.695



−0.02731−7.206−1.572−3.0678.4





Fitted second-order state-space model:
AB**K****C****D**−7.625 × $10^{-7}$
4.753 × $10^{-5}$
$5.109\;\times \;10^{-5}$−3.617 × $10^{-4}$
−27120.1262−0.09251−0.04459−3621−3.972−0.6717



Fitted first-order state-space model:
AB**K****C****D**−1.032 × $10^{-6}$
$-1.641\;\times \;10^{-7}$−0.0001816−2494−0.1363

## 5. Conclusions and Future Work

In this paper, we presented a novel tactile sensor fabricated using our robotic additive-manufacturing multiscale robotic platform NeXus. Fabrication with the NeXus reduces lengthy cleanroom processes by a factor of 10, from 10 h to less than 2. The sensor has a distinctive “fingerprint” design which allows it to be manufactured in a compact footprint of 3.8 mm × 3.8 mm. The sensor contains silver electrodes and an organic polymer film, PEDOT:PSS, which is printed in successive layers. Two layers of silicone elastomer with precise protrusions and inner chambers surround the sensor array in order to improve the consistency and detection resolution of the center of force. We performed numerous characterization tests on the tactile sensor to fully assess its capabilities and it compared with previous version made through photolithographic techniques in the cleanroom as shown in Table 5. To evaluate durability and repeatability, these tests employ a single force, ladder force, and several indentation cycles totaling more than 180,000 cycles. The sensor’s capacity to record fine spatial details is shown by the results, which have an average spatial resolution of 827 microns. Additionally, the sensor displays an average sensitivity of 1.795 µV/N in the 0.5–2 N force range, demonstrating its capacity to successfully detect minute force fluctuations. The study included a system identification analysis to obtain additional insights into the sensor’s behavior and reaction to external stimuli.

The dynamic relationship between the applied forces and the sensor’s output was described by this analysis, showing a time constant of 63 ms. The information about the dynamic response is crucial for improving the sensor’s performance in dynamic situations and adjusting to changing forces over time. In connection with Skinsim, a simulation program created to simulate large sensor arrays for robot applications, the system identification of the tactile sensor offers its model for next-level applications [63].

The proposed tactile sensor has significant promise for use in a wide range of fields, especially in robotics, prosthetics, and human–machine interfaces. It is a desirable option for situations requiring precise tactile feedback and force sensing due to its high spatial resolution, sensitivity, and dynamic nature. The discovery paves the way for more complex and natural human–machine interactions, enhancing user experience across a range of fields. While this paper describes the astonishing possibilities of the original material sensor, there are a few opportunities for future investigations. Addressing difficulties connected with versatility, large-scale manufacturing, and cost viability through the use of direct-writing inkjet printing technology could prompt a more extensive reception and arrangement of this innovation in certifiable applications. Additionally, the sensor’s capabilities could be further validated, and its potential applications expanded by examining its performance in dynamic and complex environments.

## Figures and Tables

**Figure 1 sensors-23-08280-f001:**
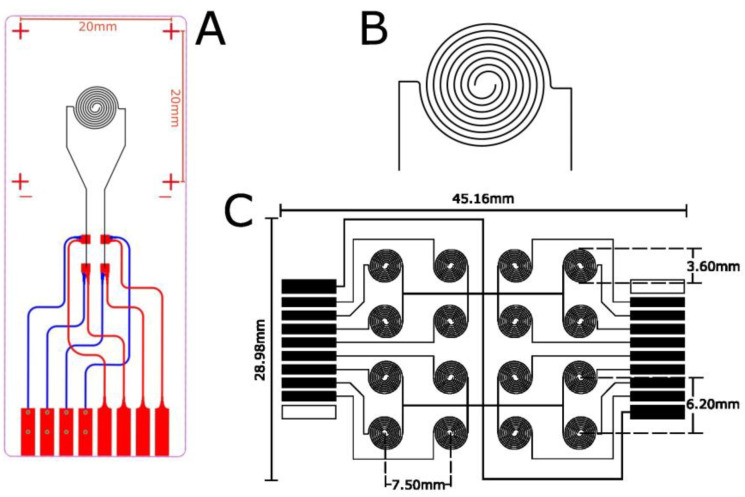
(**A**) Single fingerprint tactile on a flexible substate; (**B**) single tactile sensor; (**C**) 4 × 4 sensor array.

**Figure 2 sensors-23-08280-f002:**
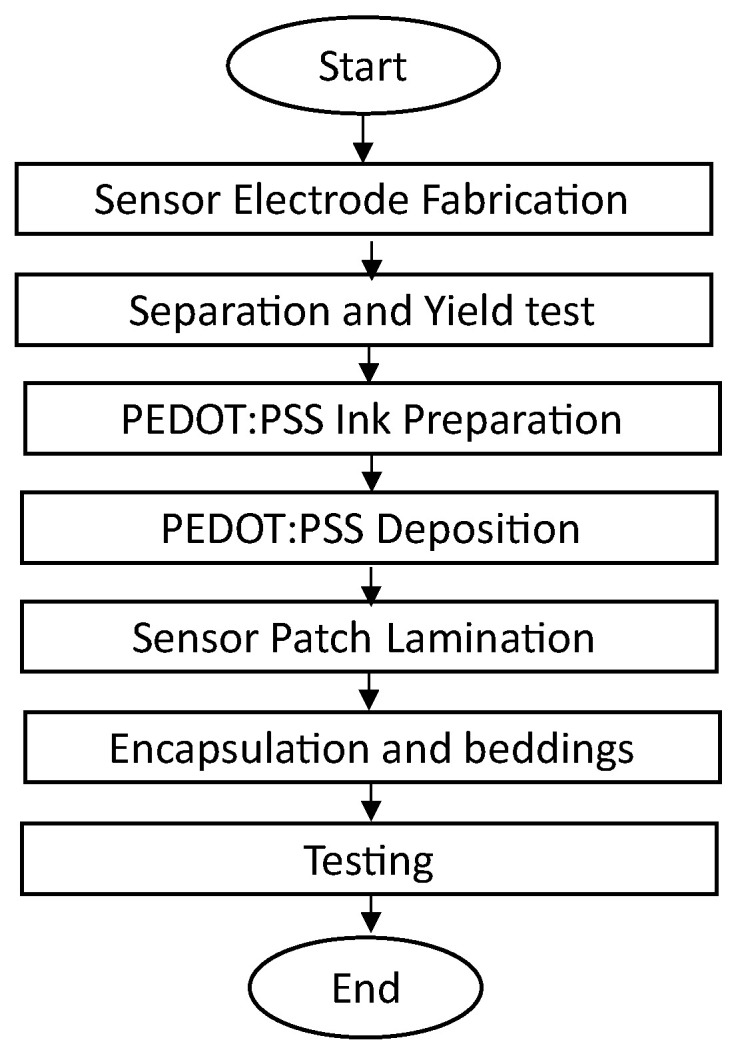
Tactile sensor fabrication flowchart.

**Figure 3 sensors-23-08280-f003:**
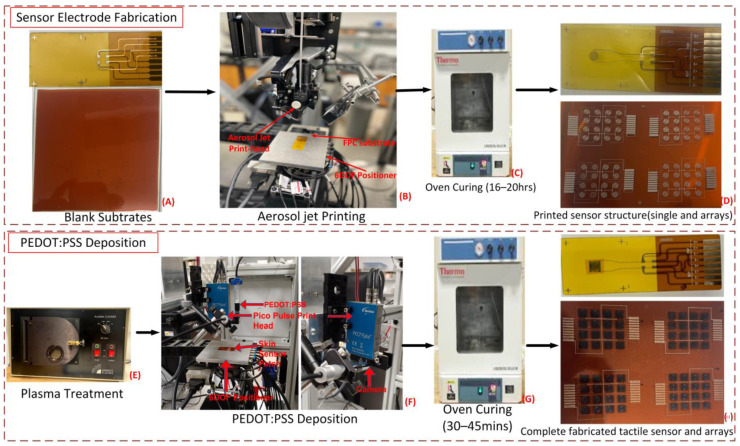
Fabrication process of a single tactile sensor and sensor patch arrays.

**Figure 4 sensors-23-08280-f004:**
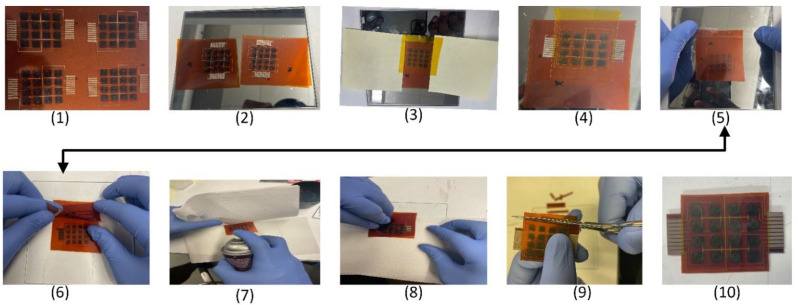
Lamination process for the double-layer skin sensor array.

**Figure 5 sensors-23-08280-f005:**
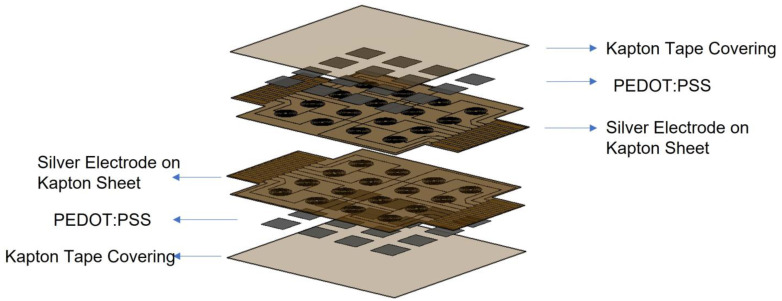
Overview of the layer design of a 4 × 4 SkinCell sensor array.

**Figure 6 sensors-23-08280-f006:**
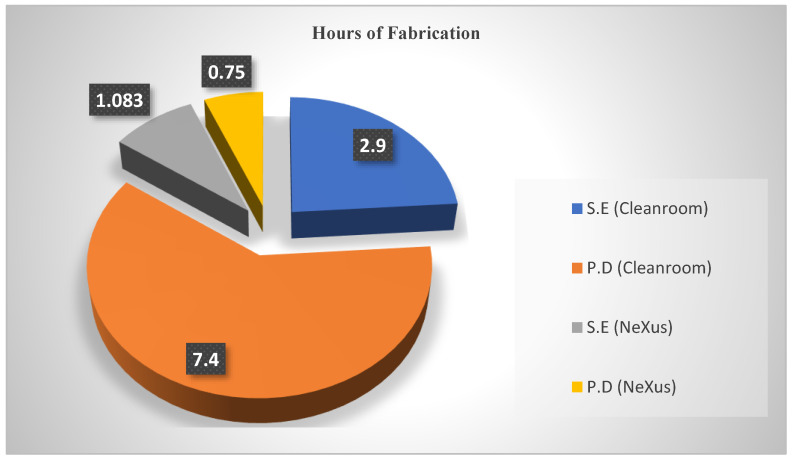
An analysis of the duration of fabrication for the cleanroom and NeXus inkjet printing fabrication process, where S.E represents the patterning of sensor electrodes and P.D is the deposition of PEDOT:PSS.

**Figure 7 sensors-23-08280-f007:**
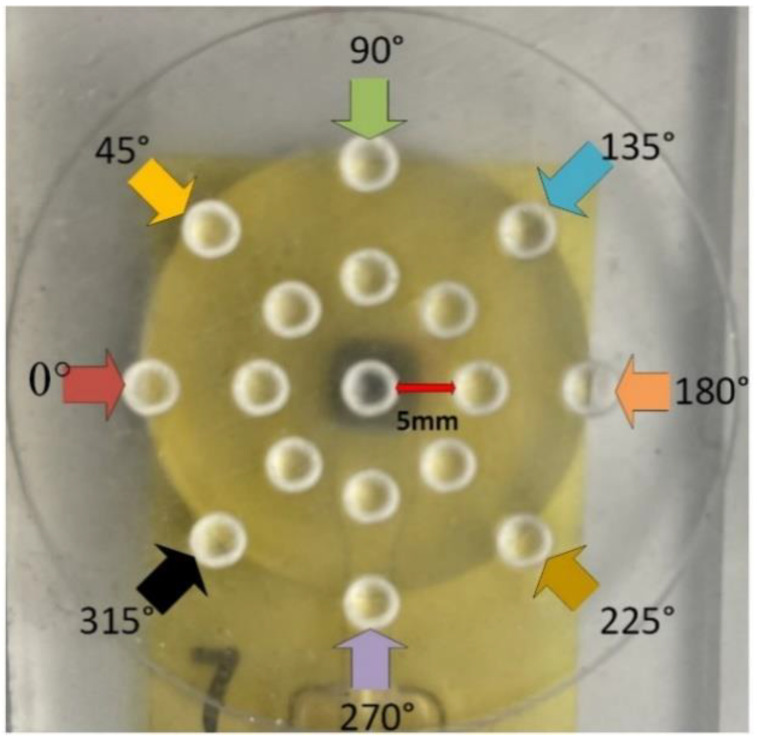
Characterization of a tactile sensor printed on a Kapton^®^ substrate with 8 separate sub-indenter locations marked for orientations of 0°, 45°, 90°, 135°, 180°, 225°,270°, and 315°.

**Figure 8 sensors-23-08280-f008:**
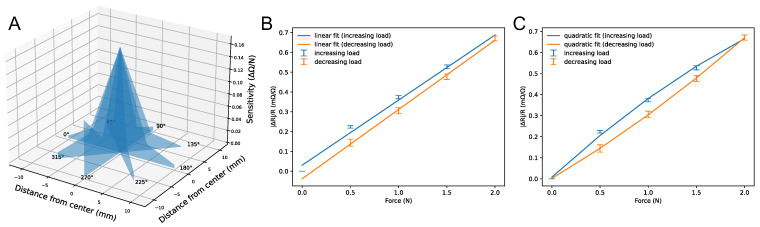
The relationship between resistance and the force applied to the sensor (0–2 N). (**A**) The magnitude of the force response from various locations, indicating a Gaussian-shaped response with a peak at the center and almost no response as we move further from the center; (**B**) the absolute linear fit of the sensitivity with respect to the applied load; (**C**) a quadratic fit describing the hysteresis of the tactile sensors based on the loading and unloading of an applied force.

**Figure 9 sensors-23-08280-f009:**
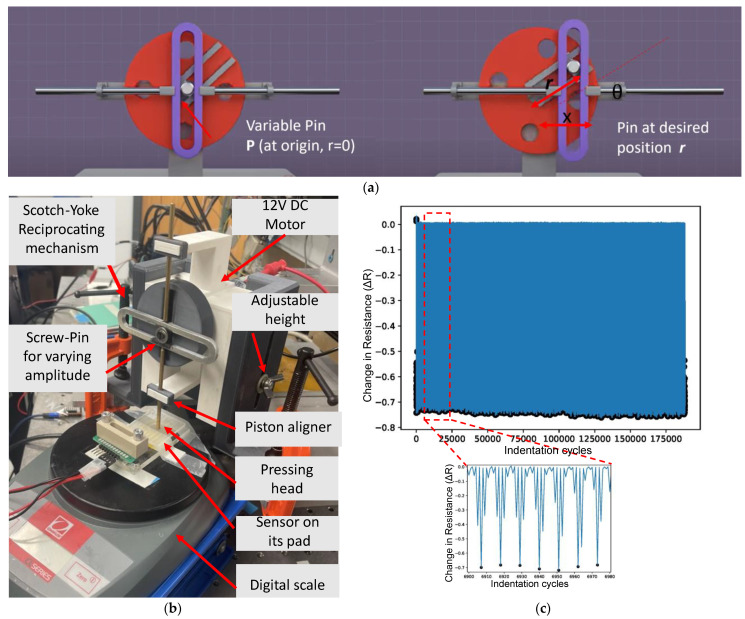
(**a**) Electronic schematic of the test setup; (**b**) Durability and repeatability test setup; (**c**) durability test profile for more than 180,000 cycles of indentation.

**Figure 10 sensors-23-08280-f010:**
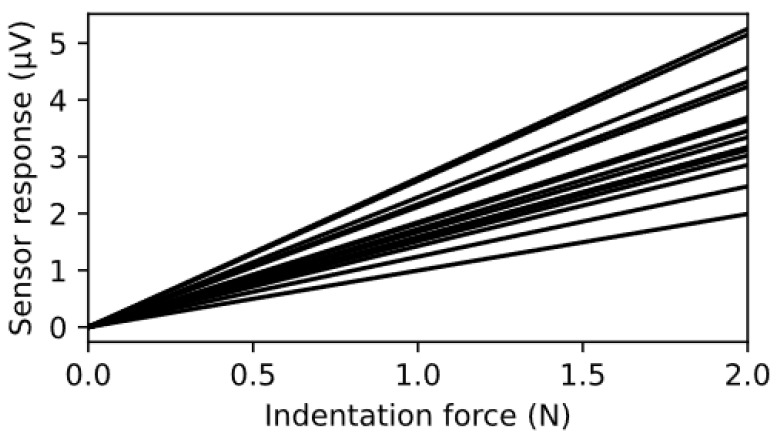
The indentation responses of all 16 tactels to an increased load, as measured via a conditioning electronic circuit.

**Figure 11 sensors-23-08280-f011:**
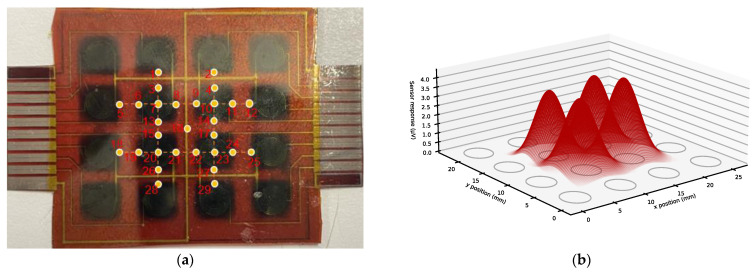
(**a**) A sensor’s response to 2 Newtons of force distributed along the *x* and *y* axes determines its spatial resolution; the number colored circle represents the force load application points (**b**) Gaussian curve fits of the four centered tactile sensor responses.

**Figure 12 sensors-23-08280-f012:**
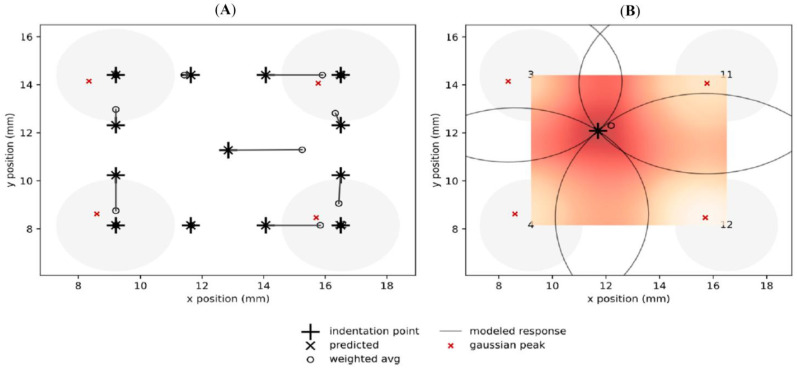
(**A**) Using the measured data from the sensor patch, the weighted average was compared with the predicted response of the elliptical model; (**B**) the intersecting ellipses predict the point of indentation in comparison with a weighted average.

**Figure 13 sensors-23-08280-f013:**
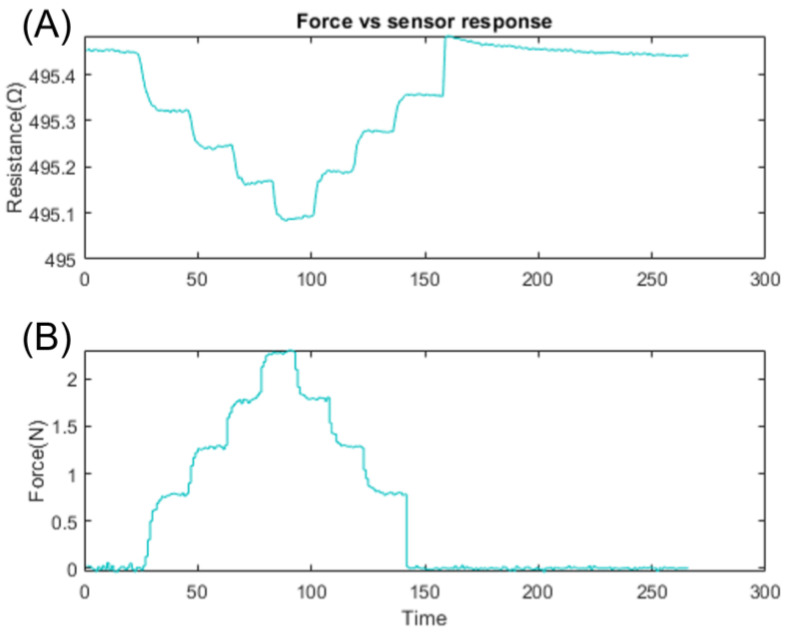
Step ladder response of the tactile sensor to the change in applied force (0–2 N), (**A**) the change in the resistance of the sensor, and (**B**) an applied force ranging from 0 to 2 N.

**Figure 14 sensors-23-08280-f014:**
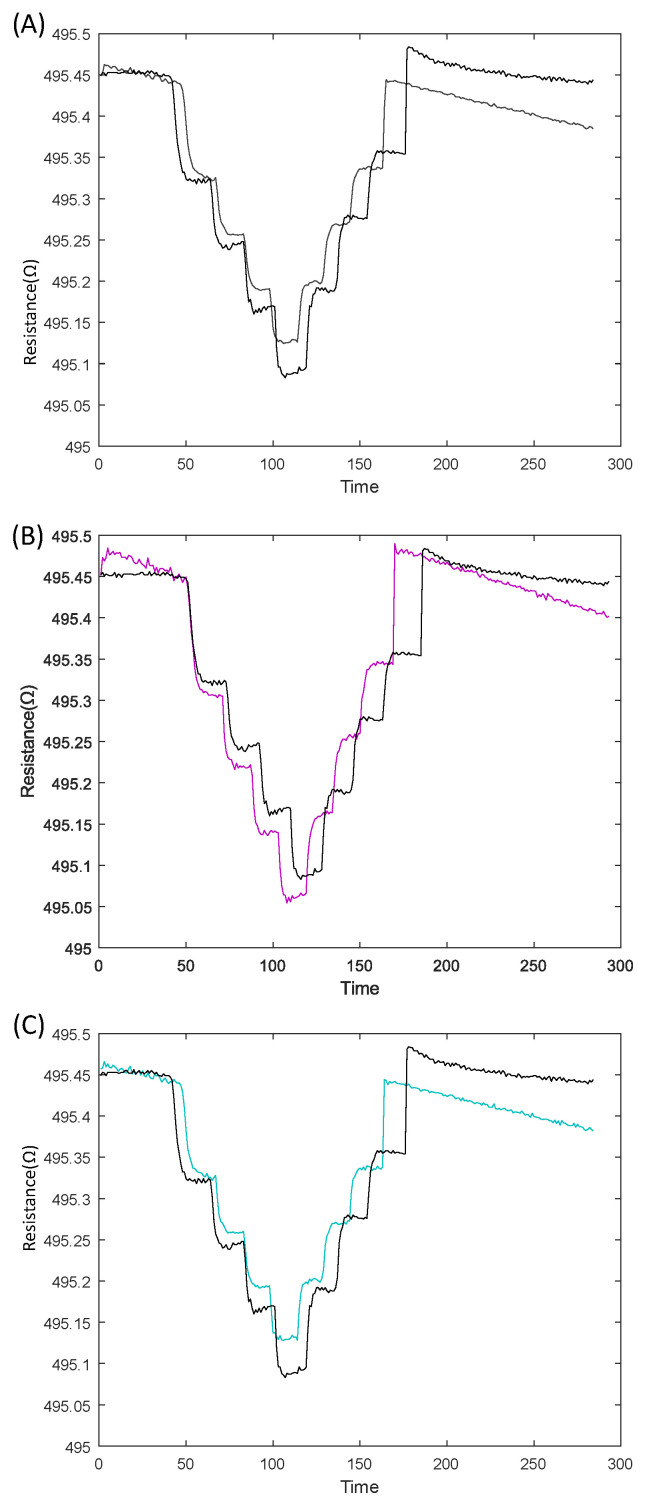
Model outputs from simulations of the (**A**) third-order model, (**B**) second-order model, (**C**) and first-order model, with validation data fit values of 65.1%, 63.02, and 63.92, respectively. The black lines represent the system output while the colored line the model response.

**Table 1 sensors-23-08280-t001:** Parameters for aerosol jet printing.

Sheath Flow Rate	135 sccm	Print Speed	10 mm/s
Atomizer Flow Rate	15 sccm	Atomizer Bath Temperature	27 °C
Atomizer Current	400 mA	Stand-off Distance	3 mm

**Table 2 sensors-23-08280-t002:** Parameters for PEDOT:PPS deposition.

$f_{d}$ [Hz]	Stroke	$P_{a}$ [psi]	$T_{f}$ [°C]	h [mm]	$\bar{d_{d}}$ (μm)	$S_{d}$ (μm)
3.3	80%	20	40	3	400	10

**Table 3 sensors-23-08280-t003:** Resistance measurements of two paired sensor patches.

Number of Sensor	Resistance (Ω)	Number of Sensor	Resistance (Ω)
1	159	1	184
2	145	2	169
3	137	3	160
4	142	4	165
5	122	5	148
6	121	6	152
7	128	7	158
8	136	8	167
1	151	1	151
2	133	2	145
3	130	3	135
4	146	4	134
5	141	5	120
6	124	6	117
7	141	7	115
8	165	8	124

**Table 4 sensors-23-08280-t004:** Estimated parameters For the Gaussian model in Figure 10.

SkinCell	$R^{2}$	Height	$\mu_{x}$	$\mu_{y}$	$\sigma_{x}$	$\sigma_{y}$	Offset (mm)
1	0.966	357	8.35	14.1	2.08	1.64	0.887
2	0.964	384	15.8	14.1	2.19	1.72	0.804
3	0.978	371	8.60	8.62	2.17	1.92	0.761
4	0.958	429	15.7	8.47	2.14	1.97	0.855

**Table 5 sensors-23-08280-t005:** Comparison between the tactile sensors fabricated in the cleanroom [41,42,43] and with the help of Nexus system.

Comparison	Cleanroom Sensors	Fingerprint Sensors
Fabrication	<10 h	<1 h
Yield	Average, 75%	100%
Ease of manufacture/ Fab. Process complexity	Complex	Simplified
Substrate Geometry/ Surface Area	Limited by silicon wafer	Custom
Reliability (indentation cycles)	>100	180,000
Durability	Wearing off terminals (fragile metal electrodes)	Printed electrodes are more resistant to wear
Spatial Resolution	N/A	>1 mm
Cost	Average (USD 2172)	Average (USD 60)

N/A—not available.

## Data Availability

All relevant data and information can be found in the paper.

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
