# Peer review of "Design, Fabrication, and Characterization of Inkjet-Printed Organic Piezoresistive Tactile Sensor on Flexible Substrate"

_sensors, 2023, doi:10.3390/s23198280_

Round 1

Reviewer 1 Report

This paper proposes a tactile sensor using a fingerprint-type (spiral) printed pattern.

Although tactile sensors using printed electronics are not considered novel, aerosol inkjet is an interesting technology and the ingenious mechanical structure sandwiched between films is attractive. The description of the experiments in this paper is detailed and provides the reader with enough information to evaluate this technique. The reviewer especially appreciates the fact that it has even been evaluated for durability. This paper provides the reader with information about a sensor with a novel structure and serves well to help the reader evaluate and further develop it technically.

While not a modification request, the introduction is a bit too long and would like something brief.

Author Response

For research article

Response to Reviewer 1 Comments

Manuscript Title: Design, Fabrication, and Characterization of Inkjet-Printed Organic Piezoresistive Tactile Sensor on Flexible Substrate

Authors: Olalekan O. Olowo*, Bryan Harris, Daniel Sills, Ruoshi Zhang, Andriy Sherehiy, Alireza Tofangchi, Danming Wei and Dan O. Popa

Summary:

Thank you very much for taking the time to review this manuscript. We thank the reviewer for their comments and feedback on our first submission. We have made numerous changes to the manuscript and organized it well.

Comment 1 - While not a modification request, the introduction is a bit too long and would like something brief.

Response 1:

Thank for your comment. The introduction is quite lengthy as a lot of previous work has been done in the field of tactile skins for robotics. To improve clarity, we have reorganized the Introduction into three sub-sections, one discussing Tactile Sensors for Robots, one discussing sensor requirements and modes of transduction in past work, and the last one describing our paper’s contributions.

Reviewer 2 Report

Journal Name: Sensors-MDPI

Title:  Design, Fabrication, and Characterization of Inkjet-Printed Organic Piezoresistive Tactile Sensor on Flexible Substrate

Summary: 

The "SkinCell," a novel tactile sensor of 3.80 mm x 3.80 mm in a spiral pattern, is introduced in the paper. The sensor is arranged in a 4x4 array to form a pad that is roughly 45mm by 29mm in size. The SkinCell is made using a specialized microfabrication platform called NeXus, which uses inkjet printers to deposit organic piezo-resistive polymer (PEDOT:PSS) and conductive silver ink onto the sensor. With two sensor sheets arranged in a half-wheatstone-bridge configuration to improve sensitivity and temperature compensation, the resulting structure is sintered and laminated. Layers of silicone elastomer with particular characteristics to improve detection resolution surround the sensor array. The results of the SkinCell's characterization are also included in the paper. They show its spatial resolution, sensitivity, and dynamic reaction time, which make it appropriate for use in a variety of applications, such as human-robot interaction.

This work is interesting and does not matches the scope of MDPI sensors. However, the review article needs to be improved further to reach the general audience of MDPI sensors. Based on the above considerations, a minor revision is given. Here are some suggestions.

Major revision

1.      The authors should cite the articles from 2022 and 2023; kindly cite more articles of MDPI-sensors

2.      Include the expanded form of PEDOT (Poly (3,4-ethylene dioxythio-phene)-poly(styrene sulfonate)) in the abstract.

3.      How does the SkinCell tactile sensor's distinct "fingerprint" design improve its performance in comparison to other models?

4.      Can you go into further detail about the NeXus platform's fabrication process, including the utilization of inkjet printers and the materials employed?

5.      What are the benefits and how does it affect sensitivity and temperature compensation when the sensor sheets are arranged in a half-Wheatstone-bridge configuration?

6.      Could you elaborate on the silicone elastomer layers' unique characteristics and how they improve detection resolution?

7.      What were the main conclusions about the performance and endurance of the sensor from the static and dynamic force loading tests?

8.      What are the real-world uses for which the SkinCell tactile sensor is particularly well suited given its properties of spatial resolution, sensitivity, and reaction time?

Are there any restrictions or difficulties that came up during the design or testing of the SkinCell sensor that ought to be taken into account for upcoming advancements or future research directions?

Author Response

For research article

Response to Reviewer 2 Comments

Manuscript Title: Design, Fabrication, and Characterization of Inkjet-Printed Organic Piezoresistive Tactile Sensor on Flexible Substrate

Authors: Olalekan O. Olowo*, Bryan Harris, Daniel Sills, Ruoshi Zhang, Andriy Sherehiy, Alireza Tofangchi, Danming Wei and Dan O. Popa

Summary:

Thank you very much for taking the time to review this manuscript. We thank the reviewer for their comments and feedback on our first submission. We have made numerous changes to the manuscript and organized it well to address your comments. Please find the detailed responses below.

Comment1: The authors should cite the articles from 2022 and 2023;kindly cite more articles of MDPI-sensors.

Response 1:

Thank you for your suggestion, a few relevant articles have been added:

  1. Mandil, V. Rajendran, K. Nazari, and A. Ghalamzan-Esfahani, "Tactile-Sensing Technologies: Trends, Challenges and Outlook in Agri-Food Manipulation," Sensors, vol. 23, no. 17, p. 7362, 2023

Lu, G., Fu, S., Zhu, T., & Xu, Y. (2023). Research on Finger Pressure Tactile Sensor with Square Hole  Structure Based on Fiber Bragg Grating. Sensors, 23(15), 6897.

Nguyen, V.-C., Oliva-Torres, V., Bernadet, S., Rival, G., Richard, C., Capsal, J.-F., Cottinet, P.-J., & Le, M.-Q. (2023). Haptic Feedback Device Using 3D-Printed Flexible, Multilayered Piezoelectric Coating for In-Car Touchscreen Interface. Micromachines, 14(8), 1553.

Feng, W., Li, P., Zhang, H., Sun, K., Li, W., Wang, J., Yang, H., & Li, X. (2023). Silicon Micromachined TSVs for Backside Interconnection of Ultra-Small Pressure Sensors. Micromachines, 14(7), 1448.

Comment 2: Include the expanded form of PEDOT (Poly (3,4-ethylenedioxythio-phene)-poly(styrene sulfonate)) in the abstract.

Response 2: The expanded form of PEDOT:PSS was included in the abstract of the manuscript upon crosscheck. This can be verified in line 14 of the manuscript.

Comment 3: How does the SkinCell tactile sensor's distinct "fingerprint" design improve its performance in comparison to other models?

Response 3: The Fingerprint sensor array design uses inkjet printing technology and can be printed in at constant substrate velocity, resulting in uniform line widths and a nearly 100% fabrication yield when compared to other models fabricated in the cleanroom [47,48].  In comparison, to previous models fabricated using cleanroom methods, our design is more durable to wear and tear resisting peeling off in contact with electronic ZIF connectors Furthermore, a spiral geometry utilizes space on the substrate in a very compact manner, basically making very compact interdigitated electrodes (IDE) that have circular symmetry. In comparison with other IDE electrode designs, such as the one in [53] it improves sensitivity by a factor of 3.

A paragraph clarifying this has been added at the end of Section 2 in lines 178-180.

Comment 4: Can you go into further detail about the NeXus platform’s fabrication process, including the utilization of inkjet printers and the materials employed?

Response 4:

NeXus is a custom multi-scale robotic system integrating automated assembly and additive manufacturing tools that we have designed and integrated in our lab. It combines several subsystems, such as industrial robotic arms, a custom 6-DOF positioner, an Optomec Decathlon Aerosol Inkjet printer, a 3D FDM printer, a Nordson EFD PicoPulse® inkjet station, and an intense pulse light (IPL) sintering station from Xenon Corp. It also includes a micro-assembly/inspection station, an e-textile loom/weaving instrument, and an industrial robot tool changer for various automated tasks. The NeXus system uses OPTOMEC® Aerosol Inkjet printer to print the sensor structure of the SkinCell tactile sensor on the flexible Kapton® substrate and uses the PicoPulse® for the deposition of the PEDOT: PSS the organic polymer responsible for the piezoresistive phenomenon of the tactile sensor. To carry the substrate to respective stations for the printing of the sensor electrode and deposition of PEDOT:PSS, a custom 6-DOF positioner in the NeXus has been designed and contains a long linear stage and 5 high-precision motorized stages. The long linear stage takes the sample between different printing, curing, and metrology instruments, while the precision motorized stages ensure XYZ displacement with 1 micron resolution as well as sample rotation and tilt. Aerosol and inkjet printing processes in the Nexus are preceded by substrate calibration and alignment which enables precise motion control and deposition of the ink at desired location.  Once the alignment is completed, the printing starts, and after the sensor electrode is printed and cured the structure is plasma treated using IPL. The plasma treated substrate is returned to the sample chuck and the moved along the long linear stage to the PicoPulse® deposition station for the deposition of PEDOT:PSS.

A paragraph adding more details to the printing process was added in Section 3.1 of the manuscript.

Comment 5: What are the benefits and how does it affect sensitivity and temperature compensation when the sensor sheets are arranged in a half-Wheatstone-bridge configuration?

Response 5:

When two strain gauge sensors are arranged in a half-Wheatstone-bridge configuration, there are several benefits and effects on sensitivity and temperature compensation:

Benefits:

  1. Sensitivity Enhancement: One of the primary advantages of using a half-Wheatstone-bridge configuration is increased sensitivity. By connecting two strain gauges in this manner, you effectively double the strain-induced resistance change compared to using a single strain gauge.
  2. Improved Linearity: The Wheatstone bridge configuration inherently provides linear output in response to strain or pressure variations. This is crucial for accurate measurements, as it allows for a direct relationship between the output signal and the applied force or deformation.
  3. Reduced Temperature Effects: A half-bridge configuration provides some degree of temperature compensation. By connecting two strain gauges with opposite sensitivity to a common Wheatstone bridge, temperature-induced changes in resistance tend to cancel each other out. This can help mitigate the effects of temperature variations on the sensor's accuracy.

A sentence was added to clarify the benefits of this configuration in the paper abstract line 17, and also  at the end of Section 2 and section 4.3.

Comment 6: Could you elaborate on the silicone elastomer layers' unique characteristics and how they improve detection resolution?

Response 6:

The silicone elastomer cavity under each tactel was proposed to increase the amount of strain seen by each SkinCell tactel. And a silicone elastomer indenter layer was also placed on top of each tactel to concentrate stress on individual tactels.

The bedding design and analysis was presented in detail in another publication [Zhang, R., Lin, J.-T., Olowo, O. O., Goulet, B. P., Harris, B., & Popa, D. O. (2022). SkinCell: A Modular Tactile Sensor Patch for Physical Human–Robot Interaction. IEEE Sensors Journal, 23(3), 2833-2846].

A small paragraph clarifying the benefits of this configuration was added in Section 4.3, lines 412-416 of the manuscript.

Comment 7: What were the main conclusions about the performance and endurance of the sensor from the static and dynamic force loading tests?

Response 7:

The conclusion of the performance and endurance of the sensor was re-stated and clarified in the Conclusion, lines 566-576. By carrying out testing on our sensor arrays, we confirmed that they can be used in robotic skin applications. Several characterization tests used single force, ladder force, and multiple indentation cycles totaling more than 180,000 cycles to assess durability and repeatability. The sensor array has under millimeter spatial resolution of 827 microns, demonstrating the sensor's ability to resolve tiny spatial features. Additionally, in the 0.5-2N force range, the sensor has an average sensitivity of 1.795 V/N, proving its capability to accurately detect minute force variations. System identification analysis determined a very responsive time constant of 63 ms, describes the dynamic relationship between applied forces and sensor output.

Comment 8: What are the real-world uses for which the SkinCell tactile sensor is particularly well suited given its properties of spatial resolution, sensitivity, and reaction time?

Response 8:

The suggested tactile sensor holds great promise for applications in many different industries, particularly in robotics, prosthetics, and human-machine interfaces. Due to its high spatial resolution, sensitivity, and dynamic nature, it is a desirable alternative for scenarios requiring precise tactile feedback and force sensing. We added a paragraph in the conclusion, lines 587-593 to better explain this.

Are there any restrictions or difficulties that came up during the design or testing of the SkinCell sensor that ought to be taken into account for upcoming advancements or future research directions?\

Response 9

One of the current restrictions is the size of sample chuck of 5(?) inches where the Kapton substrate is placed, which limits the number of sensors that can be printed on a single sheet

.

Reviewer 3 Report

The authors presented a novel tactile sensor with a “fingerprint” design that exhibits good durability and repeatability. However, the manuscript still has some problems that need to be revised.

1.       The author should calculate the curing time after printing the electrodes when analyzing the process time; after all, the electrodes prepared by sputtering and etching processes do not require a long curing time. 

2.       The experimental details in [y] mentioned by the authors in line 354 is confusing. 

3.       The author should have given a brief description about the principle of the sensor, and compare your own results explicitly with what has been published in the past literature (in a table, for instance) and to show the benefit(s) and shortcomings of their device.

 1.       The author should use punctuation correctly. For example, the punctuation in lines 194 and 306 is irregular. 

Author Response

For research article

Response to Reviewer 3 Comments

Manuscript Title: Design, Fabrication, and Characterization of Inkjet-Printed Organic Piezoresistive Tactile Sensor on Flexible Substrate

Authors: Olalekan O. Olowo*, Bryan Harris, Daniel Sills, Ruoshi Zhang, Andriy Sherehiy, Alireza Tofangchi, Danming Wei and Dan O. Popa

Summary:

Thank you very much for taking the time to review this manuscript. We thank the reviewer for their comments and feedback on our first submission. We have made numerous changes to the manuscript and organized it well to address your comments. Please find the detailed responses below.

Reviewer 3

The authors presented a novel tactile sensor with a “fingerprint” design that exhibits good durability and repeatability. However, the manuscript still has some problems that need to be revised.

Comment 1. The author should calculate the curing time after printing the electrodes when analyzing the process time; after all, the electrodes prepared by sputtering and etching processes do not require a long curing time.

Response 1:

As rightly stated, sputtering and etching processes do not require a long curing time while the curing time for the printed electrode in the oven could be from 30 minutes to 15 hours depending on the predetermined characteristics of the tactile sensor needed. Also, the curing time in the oven could be reduced to a matter of seconds using the IPL (intense pulse light) presented within the Nexus fabrication platform.

We made a clarifying statement about this in Section 3.6 in lines 333-335.

Comment 2. The experimental details in [y] mentioned by the authors in line 354 is confusing.

Response 2:

Thank you for your keen observation, it has been duly addressed in the manuscript in line 401.

Comment 3. The author should have given a brief description about the principle of the sensor and compare your own results explicitly with what has been published in the past literature (in a table, for instance) and to show the benefit(s) and shortcomings of their device.

Response 3:

The principle of the sensor is a based on the piezoresistivity of the PEDOT:PSS organic polymer changing its electric properties under the strain due to applied force. The  PEDOT:PSS is applied connecting the open circuit patterned fingerprint tactile sensors laminated back-to-back to form a half Wheatstone bridge circuit eliminating thermal and temperature drift also preventing it been exposed to moisture.

The description of the sensor’s operation principle is added to the chapter 1, section 1.3, page 3, line 103-128.

The table below shows the comparison between the tactile sensors fabricated in the cleanroom [41-43] and with the help of Nexus system. Table is added to manuscript in chapter 5, page 19, line 581.

Comparison

Cleanroom Sensors

Fingerprint sensors

Fabrication

<10hours

<1hours

Yield

Averagely 75%

100%

Ease of manufacture/

Fab. Process complexity

complex

simplified

Substrate - Geometry/

 Surface  Area

limited by silicon wafer

Custom

Reliability

(indentation cycles)

>100

180,000

Durability

wearing off terminals

(fragile metal electrodes)

printed electrodes more resistant to the wear

Spatial Resolution

N/A

>1mm

Cost

Averagely ($2172)

Averagely ($60)

Reviewer 4 Report

Interesting paper addressing the Design, Fabrication, and Characterization of Inkjet-Printed Organic Piezoresistive Tactile Sensor on Flexible Substrate

The paper in my opinion is a well structured work with analytical work and conclusions supported by experimental data.

I have some questions and suggestions for paper improvement:

1-For the sake of completeness I think is worth to include an explanation of the piezoresistivity of the PEDOT:PSS material.

2-It is known that PEDOT:PSS material is sensible to strain, temperature and humidity, please introduce the necessary actions to compensate these effects.

3-Every piezoresistive sensor displays an electromechanical gain coefficient known as strain gage, for the present sensor please add the gage factor obtained for the deposited PEDOT:PSS film.

4-Thickness of deposited PEDOT:PSS film  was measured? If yes, include the data in the text.

5- PEDOT:PSS film behavior with various thicknesses was investigated?

6- In the case of the SkinCell 4x4 sensor array please add more information (elasticity, thermal conduction) on contact glue for back to back kapton adhesion, 

7- In 354: A customized electronics circuit is used in the experiment, which  is explained in detail in [y]. Please correct and include the correct reference.

8-It would be interesting to insert an electronic schematic for the experimental test of  durability and repeatability.

9- Indentation responses of the 16 tactels has a large standard deviation; besides the different value of sensor resistance.  Do you think there is another influence due to sensor  back to back bonding?

1-Please double check your text for English mistyping errors.

2- Please check for  consistency of  bibliography (see ref 43, and correct)

Author Response

For research article

Response to Reviewer 3 Comments

Manuscript Title: Design, Fabrication, and Characterization of Inkjet-Printed Organic Piezoresistive Tactile Sensor on Flexible Substrate

Authors: Olalekan O. Olowo*, Bryan Harris, Daniel Sills, Ruoshi Zhang, Andriy Sherehiy, Alireza Tofangchi, Danming Wei and Dan O. Popa

Summary:

Thank you very much for taking the time to review this manuscript. We thank the reviewer for their comments and feedback on our first submission. We have made numerous changes to the manuscript and organized it well to address your comments. Please find the detailed responses below.

Comment 1-For the sake of completeness I think is worth to include an explanation of the piezoresistivity of the PEDOT:PSS material.

Response 1:

The explanation of the Piezoresistivity of the PEDOT:PSS material is given below ( as seen in line 109 to 120 of the manuscript):

Piezoresistivity in PEDOT:PSS, a conductive polymer composite, is a complex phenomenon  where its electrical resistance changes in response to mechanical deformation. This  most likely  occurs due two main effects: a first one, more robust process, when the material is subjected to stress or deformation, such as bending or compression, the spacing and alignment of its conducting polymer chains (PEDOT) are altered. This, in turn, affects the material's electrical conductivity. When chains are closer, electrical resistance decreases, and when they're pushed apart, resistance Increases. A second effect is related to the changes of the electronic structure of the PEDOT and PSS due to the mechanical deformation, affecting the polymer chains, bonding distances between the atoms, and in a result charge distribution. It can be speculated that this causes changes of the energy difference between occupied and unoccupied electron states in PEDOT and PSS. By analogy to the piezoresistive effect in the bulk solid semiconducting materials, where beside geometrical (volume) part affecting resistivity, there is also component related to the changes in electron structure of the material -  inducing changes of the band gap value (for example in case of SI).

Comment 2-It is known that PEDOT:PSS material is sensible to strain, temperature, and humidity, please introduce the necessary actions to compensate these effects.

Response 2:

As rightly stated, temperature and humidity have effects on PEDOT:PSS, but strain is required to determine its response to applied force. To address the issue of humidity, lamination was done to seal off the surface of the sensor prevent humidity from getting in as described in section 3 of the manuscript. To compensate for the temperature a half-Wheatstone-bridge configuration consisting of two tactile sensor patches were assembled together. By connecting two strain gauges with opposite sensitivity to a common Wheatstone bridge, temperature-induced changes in resistance tend to cancel each other out to some extent. This can help mitigate the effects of temperature variations on the sensor's accuracy. (This is discussed in lines 262-265, 419-420).

Comment 3-Every piezoresistive sensor displays an electromechanical gain coefficient known as strain gage, for the present sensor please add the gage factor obtained for the deposited PEDOT:PSS film.

Response 3:

Typically, PEDOT:PSS has a gauge factor in the range of 6.9 to 17.8. [50,51]. We have not directly measured the gauge factor for this particular composition of our PEDOT:PSS samples, but we fully expect they be in this range. A clarifying statement was added in Section 1.3 lines 120-122.

Comment 4-Thickness of deposited PEDOT:PSS film was measured? If yes, include the data in the text.

Response 4:

Yes, PEDOT:PSS deposited using PicoPulse® is approximately 300-400 nm  for a single layer deposited and around 1 micron  for 3-layer deposition. We have added information about thickness of the pedot:pss film measured using Dektak® Profilometer in the manuscript, chapter 3, section 3.4, line 249-250.

Comment 5- PEDOT:PSS film behavior with various thicknesses was investigated?

Response 5:

In this work, we have studied behavior of PEDOT:PSS thin films with thicknesses not exceeding 1 micron, which corresponds to 1, 2, and 3 printed layers. It was observed that a sensor with 3 layers produced the most reliable results, whereas a single layer sensor’s behavior was inconsistent. We have not studied responses of the sensor with a larger number of layers, and thicknesses above 1 micron for a given type of PEDOT:PSS ink (from Heraeus). As our goal was to demonstrate capability of fabricating as thin as possible PEDOT:PSS thin films using inkjet printing method, producing structures with acceptable behavior for given application, and reasonable manufacturing times - compared to cleanroom fabrication.

In our previous studies we have analyzed inkjet printed PEDOT:PSS films with larger thicknesses – around 15 microns. However, in those studied we have used for printing ink of different formulation – a composite of PEDOT:PSS, DMSO (Dimethyl sulfoxide), and PVP (Polyvinylpyrrolidone), with thickness approximately 15 microns [44]:

  1. Olowo et al., "INKJET PRINTING OF PEDOT: PSS INKS FOR ROBOTIC SKIN SENSORS," in Proceedings of the ASME 2022 17th International Manufacturing Science and Engineering Conference (MSEC2022) unpublished, West Lafayette, Indiana, USA, 2022, vol. 80989.

This is discussed in the paper in Section 3.4, page7 lines 245,251-256

Comment 6- In the case of the SkinCell 4x4 sensor array please add more information (elasticity, thermal conduction) on contact glue for back-to-back Kapton adhesion,

Response 6:

For the back-to-back Kapton alignment and adhesion, a sprayed 3M 13.8 oz. Super 77 Multipurpose Spray Adhesive is evenly sprayed in-between the pairs. It is designed to withstand a wide temperature range. It typically has good resistance to heat and cold, making it suitable for many applications. it also provides a secure bond; it is not typically considered an elastic adhesive like silicone or certain rubber-based adhesives. Its primary purpose is to create a strong and durable bond between materials.

A paragraph has been added in the paper, Section 3.5, lines 290-294.

Comment 7- In 354: A customized electronics circuit is used in the experiment, which is explained in detail in [y]. Please correct and include the correct reference.

Response 7:

Thank you for your keen observation, it has been duly addressed in the manuscript in line 401.

Comment 8-It would be interesting to insert an electronic schematic for the experimental test of durability and repeatability.

Response 8:

x

Posted below is the electronic schematic for the motion description of the reciprocal machine used for the experimental test of durability and repeatability. It included in the manuscript in lines 350-351 in figure 9a.

Variable Pin

P (at origin, r=0)

r

x

Pin at desired  position  r

Before the machine starts to move, the position of pin “P” can be set at a desired length which can vary from zero up to the maximum value of 10 mm.  This is done by a system of set screws system at the front and rear sides of the rotating disk ( not shown in the schematic for the sake of simplicity).

The speed of disc  is controlled by DC voltage of the motor( linear behaviour).

The position and velocity of the moving piston can be obtained by kinematic analysis as follows:

Where r is the set value of the pin radius and is the instantaneous angular position of the crank.

Comment 9- Indentation responses of the 16 tactels has a large standard deviation, besides the different value of sensor resistance. Do you think there is another influence due to sensor back-to-back bonding?

Response 9:

There are couple of reasons that could be responsible for the  significant standard deviation  and different value of sensor resistance observed.

  1. The varying length in the tactile electrode.
  2. The change in the printed line width observed as the print time increased.
  3. Also, quality of the PEDOT:PSS based on its preparation, preservation and application.

.
